# Time-Dependent Odorant Sensitivity Modulation in Insects

**DOI:** 10.3390/insects13040354

**Published:** 2022-04-02

**Authors:** Hao Guo, Dean P. Smith

**Affiliations:** 1State Key Laboratory of Integrated Management of Pest Insects and Rodents, Institute of Zoology, Chinese Academy of Sciences, Beijing 100101, China; 2Departments of Neuroscience and Pharmacology, University of Texas Southwestern Medical Center, 5323 Harry Hines Blvd., Dallas, TX 75390, USA; dean.smith@utsouthwestern.edu

**Keywords:** olfactory receptor neurons, desensitization, odorant receptors, sensitivity, *Drosophila melanogaster*

## Abstract

**Simple Summary:**

Insects, including blood-feeding female mosquitoes, can transmit deadly diseases, such as malaria, encephalitis, dengue, and yellow fever. Insects use olfaction to locate food sources, mates, and hosts. The nature of odorant plumes poses a challenge for insects in locating odorant sources in the environment. In order to modulate the system for the detection of fresh stimuli or changes in odorant concentrations, the olfaction system desensitizes to different concentrations and durations of stimuli. Without this ability, the chemotaxis behaviors of insects are defective. Thus, understanding how insects adjust their olfactory response dynamics to parse the chemical language of the external environment is not only a basic biology question but also has far-reaching implications for repellents and pest control.

**Abstract:**

Insects use olfaction to detect ecologically relevant chemicals in their environment. To maintain useful responses over a variety of stimuli, olfactory receptor neurons are desensitized to prolonged or high concentrations of stimuli. Depending on the timescale, the desensitization is classified as short-term, which typically spans a few seconds; or long-term, which spans from minutes to hours. Compared with the well-studied mechanisms of desensitization in vertebrate olfactory neurons, the mechanisms underlying invertebrate olfactory sensitivity regulation remain poorly understood. Recently, using a large-scale functional screen, a conserved critical receptor phosphorylation site has been identified in the model insect *Drosophila melanogaster*, providing new insight into the molecular basis of desensitization in insects. Here, we summarize the progress in this area and provide perspectives on future directions to determine the molecular mechanisms that orchestrate the desensitization in insect olfaction.

## 1. Introduction

Olfaction endows insects with various abilities to sense foods and mates as well as to avoid danger [1,2,3,4,5,6]. The main olfactory organs of insects are the antennae and maxillary palps, which are densely decorated with finger-like structures called sensilla [7,8]. Each sensillum contains the dendrites of 1–4 olfactory receptor neurons (ORNs) that are bathed in sensillum lymph [4,9]. The periphery olfactory responses involve three major gene families: odorant-binding proteins (OBPs), olfactory receptors, and odorant-degrading enzymes (ODEs) [10,11,12]. Over the last decades, the paradigm of olfactory information transduction and presentation has been intensively studied [13,14]. Hydrophilic food odorants appear able to dissolve in the sensillum lymph and directly interact with odorant receptors [15,16]. However, hydrophobic ligands including sex pheromones and some food odorants utilize extracellular odorant-binding proteins to penetrate the lymph and facilitate delivery to the receptors [17,18]. Molecules entering the sensillum lymph are bound with and carried by OBPs to the nearby olfactory receptors that are embedded in the membrane of ORNs. The transient ligand–receptor interaction activates receptors, converting the chemical signals into electrical impulses. Information encoded from one type of ORN is relayed through the axon of an ORN and converges onto one glomerulus in the antennae lobe. The integrated olfactory information is further projected along projection neurons (PNs) into the higher-order neural circuits, which results in behaviors. Insects employ two olfactory coding mechanisms, i.e., labeled line and combinatory coding, to sample the ambient odorants [19]. In combinatorial coding, an OR can respond to multiple general odorants, and a single odorant can activate multiple receptors. Through the combinatorial coding mechanism, insects can perceive many odors through a limited number of ORs. Insects mainly use the labeled line to sense pheromones [20,21] or vital odorants [22]. 

OBPs are a group of small soluble proteins (10–14 kDa) highly abundant in extracellular sensillum lymph (up to 10 mM) [23,24,25]. Most insect OBPs have six positionally conserved cysteine residuals that form three disulfide bonds, which, together with the six α helices, constitute a compact structure of OBPs that forms an odorant-binding cavity. The roles of OBPs in olfaction are a live debate. Studies have proposed OBPs as (1) carriers of odorants [25,26,27,28]; (2) buffering agents for maximal olfactory sensitivity [23]; (3) activators of olfactory receptors [18,29]. In the case of sex pheromones, OBP binding may contribute to specificity by selecting the odorants for delivery to the olfactory neurons. Whereas the pheromone-interacting OBPs, such as the PBPs in Lepidopteran insects and LUSH in *Drosophila melanogaster*, are necessary for normal responses to pheromones [18,26], the roles of general OBPs expressed in the basiconic sensilla tuned to plant odorants are poorly understood. Recently, it has been shown that the abundant OBPs in *Drosophila* basiconic sensilla did not affect the electrophysiological and behavioral responses to hydrophilic odorants when mutated [16,30,31,32,33]. Although these OBPs seem to be unnecessary for the peak response of corresponding ORNs, other OBPs were implicated in the modulation of response kinetics of ORNs, especially to hydrophobic odorants [34]. Apart from OBPs, a variety of enzymes, such as esterases, glutathione S-transferases, cytochromes, and aldehyde oxidases, were proposed as odorant-degrading enzymes in insects [10]. 

Neuronal olfactory receptors in insects consist of three major gene subfamilies: odorant receptors (ORs) [35,36,37], ionotropic receptors (IRs) [38], and gustatory receptors (GRs) [39,40]. Empirical evidence delineates that ORs expressed in neurons located in the basiconic and trichoid sensilla are primarily tuned to plant odorants and pheromones, respectively [41,42], whereas IRs are confined to coeloconic sensilla neurons and are largely sensitive to amines and acids [38,43]. Whereas vertebrate and nematode ORs are G-protein-coupled receptors [44,45], insect ORs are heptahelical ion channels with a cytoplasmic N-terminus and an extracellular C-terminus [46,47,48,49]. Recent studies revealed a surprising amount of overlap between the expression of ORs, GRs, and IRs in *D. melanogaster* [50]. In particular, GRs expressed in the olfactory system are specifically tuned to CO_2_ [39,40,51,52]. Insect ORs are thought to be heteromultimers composed of a tuning OR and a common subunit of odorant receptor coreceptor (Orco) [53,54,55]. Unlike vertebrate ORs, insect ORs are not G-protein-coupled receptors but instead are nonselective ligand-gated cation channels permeable for Ca^2+^, Na^+^, and K^+^, implying that they may use different types of intracellular machineries to regulate sensitivity [53,54]. In particular, whether G proteins and the related intracellular second messengers are involved in downstream modulatory processes remains a live debate. For instance, the knockdown of different isoforms of Gα in *Drosophila* antennae had little, if any, effect on olfaction responses [56]. Moreover, the application of the inhibitor of heterotrimeric G proteins in Sf9 and HEK293 cells expressing *Drosophila* OR43b had a negligible effect on the response of the receptor [48]. Yet, other studies found that either knockout or RNAi of Gαq impaired the olfactory responses [57]. In line with this, high activity of phospholipase C and rapid production of IP3 were detected in the ORNs that were activated by pheromones in cockroaches or moths [58,59,60]. Many studies have implicated cAMP signaling in insect olfaction [54,61,62]. It was reported that cAMP could activate the obligatory subunit of OR complex, Orco [54]. In contrast, the concentration of cGMP, not cAMP, increased in the antennal homogenate after pheromone stimulation in *Antheraea Polyphemus*, *Bombyx mori*, and *Heliothis virescens* [63,64]. These disparate findings may be caused by the different approaches used but reflect the heterogeneity in the intracellular signaling network. Recently, the structures of an Orco homomer from the parasitic fig wasp *Apocrypta bakeri* [65] and MhOR5 homomer from the jumping bristletail *Machilis hrabei* [66] were solved by single-particle cryo-electron microscopy. Both established structures are tetramers, which indicates that OR and Orco subunits form a heterotetramer. Although there is a single *Orco* gene in most insect species, the number of tuning *ORs* is greatly variable, ranging from three in the dragonfly *ladona fulva* [67] to hundreds in eusocial insects, such as ants and honeybees [68,69]. Orco is highly conserved across insect lineages, and the amino-acid identity of Orco from evolutionarily distant insects is above 70% [70,71]. Orco is functionally exchangeable between insects, suggesting the functions of Orco are highly conserved in insects [71]. These features and the fact that Orco is a shared subunit make Orco a promising candidate for the modulation of insect olfactory responses [72]. 

Olfactory sensitivity, which is manifested by the number of action potentials in olfactory neurons per unit time, is the determinant of the olfactory inputs to the central nervous system [15]. In nature, insects encounter myriads of odorants, the presence of which may be transient, continuous, or superimposed [73,74]. How do insects adjust the dynamic range of olfactory responses to encode fluctuations in odor space? Neural desensitization is the major mechanism through which sensory neurons expand their response dynamic ranges to detect different concentrations of odorants and a new odorant against background odorants [75]. Here, we review the recent progress in the studies of two major desensitization forms caused by different durations of stimuli. 

## 2. Desensitization of ORNs

A hallmark of desensitization is decreased olfactory responses to the prolonged presence of odorants or repeated stimuli [76,77]. In the literature, the terms “desensitization” and “adaptation” are both used for the description of this process. For clarity and coherence in this review, we use desensitization. Variable paradigms of odorant exposure have been used to trigger desensitization, and the results indicated that the degree of desensitization depends on both the intensity and duration of odorant exposure [78]. In the last decade, multiple approaches have been developed to document the olfactory desensitization and underlying mechanisms, including electrophysiological recordings (single sensillum recordings (SSRs), electroantennography (EAG), and patch-clamp recordings) and different chemotaxis assays. Depending on the time scale, the desensitization is divided into short term, which typically happens in milliseconds to seconds; and long-term, which spans minutes to hours, even days [77,79,80,81]. ORN desensitization can intrinsically occur in the dendrites of ORNs [79] or through feedback inhibition from the downstream glomerulus inhibitory neurons, such as peptidergic and GABAergic inhibitory neurons [77,82,83,84], implying multiple mechanisms underlay olfactory desensitization in insects. Because desensitization mechanisms involving glomerulus inhibitory neurons have been intensively reviewed [77], here, we focus on the desensitization that operates in the outer dendrites of ORNs. 

### 2.1. Short-Term Desensitization 

The research of short-term desensitization mostly focuses on Lepidopteran moths and *D. melanogaster*. The high sensitivity of moths to the long-chain, unsaturated pheromone compounds is a suitable model for studying short-term desensitization. In the hawkmoth *Manduca sexta*, moths prestimulated by a 250 ms duration of the pheromone bombykal were 10-fold less sensitive to a subsequent 50 ms puff of bombykal [80]. In *Grapholita molesta*, the antennal response was attenuated to the repetitive pulse of the pheromone (*Z*)-8-dodecenyl acetate, and the degree of desensitization was dependent on the stimulation rate [85]. *Drosophila* living in a plastic vial filled with agarose and pre-exposed to isoamyl acetate or benzaldehyde for 30 s were dramatically desensitized to the odorant present [86]. The chemotaxis of the pre-exposed flies to an odorant source was scrambled, and the antennal responses were greatly reduced, with no EAG response to isoamyl acetate after 30 s pre-exposure [86]. In *Drosophila*, single sensillum recording revealed that the ab3A ORN was rapidly desensitized with 250 to 500 ms puffs of intermediate and high concentrations of ethyl acetate [87]. The desensitization degrees, which are calculated by normalization of the firing rates of desensitized states compared with that of peak response, are invariable among different concentrations of stimuli. A similar phenomenon also holds true for the ab2A, ab3B, and ab7A ORNs responding to corresponding odorants [87]. Recently, patch-clamp recordings were applied to measure the responses of single ORNs, which could faithfully recapitulate the response properties and provide an opportunity for pharmacologically manipulating the intracellular conditions to study the signal transduction network. Using this preparation, *Drosophila* OR22a-expressing neurons were rapidly desensitized during a 30 s step of their ligand, ethyl propionate [88]. Similarly, OR47a-expressing neurons were also strongly adapted to a 20 s step of pentyl acetate [88]. Notably, two representative IR-expressing neurons, the ac3A ORN and the IR84-expressing ORN, housed in ac4 were not desensitized to butyric acid and phenylacetaldehyde, respectively [88]. This implies there is a fundamental difference between ORs and IRs in the ability to desensitize to stimuli.

### 2.2. Long-Term Desensitization

In nature, insects not only encounter brief stimulation pulses but also spend a long time on food or mating, experiencing prolonged olfactory stimuli. An exemplary case is a swarm of fruit flies buzzing over a pile of ripened fruits for hours and days. For example, prestimulation with high concentrations of one plant odor, hexanol, for 1 h abolished the following behavioral response of *D. melanogaster* larvae not only to hexanol but also to some other structural analogs, such as butanol, pentanol, heptanol, octanol, and nonanol [89]. Similarly, a 10 s pulse of prestimulation with a high concentration of butanol (1:10) rendered the antennae of *D. melanogaster* completely anosmic to a 1 s pulse of a low concentration of butanol (1:100), indicating strong desensitization triggered by odorant stimuli [90]. Notably, in that experiment, even though the prestimulation was only 10 s, it took the antennae at least 7 min to recover from the desensitization [90]. At the single sensillum level, pre-exposing the palpal basiconic 1A (pb1A) neuron of *D. melanogaster* with ethyl propionate for 25 s decreased the response to subsequent peak stimulations of ethyl propionate by 40% and of ethyl acetate by 50% [91]. Additionally, a pre-exposure paradigm at timescale of hours was used to induce long-term desensitization. For example, the pre-exposure of *D. melanogaster* to 5% geranyl acetate for 2 h reduced the responses to both 0.01% and 0.1% geranyl acetate by 70% [92]. Moreover, the pre-exposure of wild-type *Drosophila* to 10% 11-*cis*-vaccenyl acetate (cVA) for 1 h caused a striking reduction in olfactory sensitivity to the same concentration of cVA [79]. Furthermore, exposure to an odorant cocktail containing 10% cVA, 10% ethyl acetate, 10% ethyl butyrate, 10% pentyl acetate, and 10% octanol resulted in global desensitization of the olfactory neurons in the antennae [79]. The process of this desensitization was quite slow, reaching a maximum after 30 min, and sensitivity was gradually restored within 2 h [79]. This time course is similar to the long-term desensitization in *C. elegans* [93]. When pre-exposed to an AWC-sensing odorant for 1 h, the nematode needed 3 h for the behavioral responses to the experienced odorants to be restored [93,94]. In addition, the moth’s pheromone detection system has received extensive attention from researchers interested in long-term desensitization. Pre-exposure of male oblique-banded leafrollers, *Choristoneura rosaceana* (Harris), to their pheromone blend, (*Z*)-11-tetradecenyl acetate (Z11-14:Ac) and its isomer, for 15 or 60 min reduced antennal responses to this blend by 55–58%, as revealed by EAG recordings. The onset of long-term desensitization occurred after 5 min pre-exposure and reached the maximum in 15 min; the response was slowly and linearly recovered in time somewhat equivalent to that of pre-exposure [95]. *Aedes aegypti* mosquitoes exhibited dramatically attenuated avoidance behavior to 20% DEET following a pre-exposure to DEET for 3 h, which could be explained by the decreased responses of ORNs to DEET revealed by EAG recordings [96]. 

## 3. Molecular Mechanism of Olfactory Desensitization

The vertebrate ORs belong to classic G-protein-coupled receptors [44]. The ligand–receptor interaction activates Gαs protein (Gαolf), which then activates adenylyl cyclase type III (AC III), resulting in the production of 3′–5′-cyclic adenosine monophosphate (cAMP) and the opening of cyclic nucleotide-gated (CNG) channels that are permeable to the cations sodium and calcium [97]. The raised intracellular level of Ca^2+^ further opens the calcium-gated chloride channel, leading to the depolarization of membrane potentials [97]. Short-term desensitization of vertebrate ORs involves (1) uncoupling of ORs from G proteins, triggered by the phosphorylation of ORs by intracellular kinases, including G-protein-coupled receptor kinase 3 and protein kinase A; (2) closing of CNG channels by the calcium-binding protein calmodulin (CaM); (3) deactivation of ACIII through phosphorylation by CaM-dependent protein kinase II (CaMKII); (4) hydrolysis of cAMP by phosphodiesterases (PDEs) activated by CaMKII [78,98]. Long-term desensitization is achieved by the internalization of odorant receptors by endocytosis [99]. Compared with the knowledge of the molecular mechanism of vertebrate olfaction desensitization, the progress of studies of insect olfaction desensitization lags. In the following sections, we summarize the results of olfaction desensitization in insects. 

### 3.1. Short-Term Desensitization

Overall, the findings on the mechanisms underlying short-term desensitization in insects are not conclusive. It is highly possible that the modulation of multiple modalities contributes to short-term desensitization. It was reported that short-term desensitization occurs at the level of signal transduction, presumably involving a negative feedback loop that decreases the affinity of the receptor for the odorants [100]. However, the identity of the molecular component(s) remains enigmatic. As described above, in vertebrate animals, the influx of calcium plays a central role in rapid olfactory desensitization as it initiates a cascade of feedback pathways to quench the intracellular signaling network. In *Drosophila*, either simple removal of extracellular Ca^2+^ or clearance of free intracellular Ca^2+^ by a Ca^2+^ chelator 1,2-bis(2-aminophenoxy)ethane-N,N,N′,N′-tetraacetic acid (BAPTA), which is equilibrated through the recording electrode into the cytosol, abolishes adaptation, suggesting the influx of Ca^2+^ controls the desensitization machinery [88]. In moths, a cGMP cascade may be involved in the desensitization to pheromones. Application of exogenous cGMP to antennal homogenate dramatically attenuated the production of IP3 that was elicited by the stimulation of pheromones [64]. The caveat is that most of the experiments were conducted with antennal homogenate, which may contain molecular components from other signaling processes and even enzymes from non-neuronal cells. In *Drosophila*, the mutant of the transient receptor potential (Trp) Ca^2+^ channel displayed defective sensory desensitization, suggesting that this channel may be a target for desensitization [86]. However, neither Trp transcripts nor proteins were detected in mature antennae [86]. Thus, how Trp channels contribute to the desensitization awaits further studies. 

Extracellular soluble proteins such as OBPs and ODEs were demonstrated to modulate olfactory responses. Genetic ablation of the single abundant OBP in ab8 sensilla, OBP28a, increased the peak response of this sensillum to 1-octanol and prolonged the response after the stimuli were removed [33]. More importantly, this mutant had a significantly higher response to the pulse stimulation of 1-octanol superimposed on the background 1-octanol than that of the wild type, a phenotype reminiscent of defective olfactory desensitization [33]. Similarly, the double mutant of OSE and OSF (OBP83b and OBP83a) displayed deactivation defects [34]. The deactivation time constant (*t*) of a mutant responding to farnesol was at least four-fold higher than that of the wild type, leading to a tonic response to a pulse stimulus [34]. Intriguingly, the deactivation defect appeared to be odorant-specific, as other odorants detected by the same receptor were unaffected [34]. Moreover, in *D. melanogaster*, the mutation of one ODE, carboxylesterase 6, led to increased olfactory firing rates and prolonged responses to cVA [101]. This enzyme was expressed in non-neuronal auxiliary cells and secreted into the sensillum lymph, where it appeared to metabolize the pheromone [101]. Because the deactivation of ORs is one of the mechanisms underlying desensitization, these findings suggest that some subsets of OBPs and ODEs can participate in the desensitization of ORNs. 

Taken together, the various reports on the mechanisms potentially involved in desensitization make it hard to establish a model that encapsulates short-term desensitization in insects. However, it is reasonable to extrapolate that both OBPs and ODEs control the availability of odorants in sensilla to dynamically regulate the olfactory response kinetics, including the desensitization, of ORNs. Simultaneously, the signaling components in the olfactory transduction pathway are subjected to negative regulation imposed by a negative feedback regulatory pathway. 

### 3.2. Long-Term Desensitization

Recently, focusing on the *Drosophila* Or67d/Orco receptor that responds to the *Drosophila* pheromone cVA, a molecular switch mediating long-term desensitization was revealed. There are five conserved candidate phosphorylation sites on the intracellular loops of Orco: two predicted phosphorylation sites on the second intercellular loop of OR67d, T262 and T263; three on the third intercellular loop of Orco, T250, S289, and T327 [19,102]. Individually mutating each of them to alanine showed that only the mutant of S289 on Orco affected the odorant sensitivity, indicating that S289 is an important phosphorylation site that is essential for odorant sensitivity modulation. Conversely, mutation of this serine to aspartic acid (Orco S289D) increased the sensitivity of olfactory neurons. These findings suggest that Orco S289 is the toggle that regulates the olfactory response’s sensitivity through phosphorylation and dephosphorylation [80]. In both cases, the mutations had no effect on the expression levels or subcellular localization of Orco in the olfactory neurons. Furthermore, odorant exposure induced dephosphorylation of Orco at S289, and phosphorylation was recovered following removal of the stimulus [77,79]. This indicated that S289 is phosphorylated under maximum sensitivity conditions, and dephosphorylation reduces sensitivity of the receptors. In vertebrate animals, one of the major principles of desensitization is the β-arrestin2-mediated internalization of ORs once they become phosphorylated by cAMP-dependent protein kinase A following a prolonged exposure [99]. In contrast, the level of Orco on dendritic cilia remained unchanged, suggesting that the slow desensitization of insect ORs does not involve receptor translocation or internalization [79]. Interestingly, this phosphorylation site is conserved across diverse insect species, including the Dipteran mosquitoes, Lepidopteran moths, ants, parasitoids, and beetles, rendering it a promising target for the manipulation of olfactory sensitivity for pest control (Figure 1). 

To identify the kinases responsible for the phosphorylation on OrcoS289, transcriptome analysis identified 21 candidate serine and threonine kinases expressed in antennae [103]. RNAi screening revealed that the knockdown of one member of the protein kinase C family, PKC98E, significantly reduced olfactory sensitivity when expressed in the olfactory neurons. Anti-PKC98E antiserum revealed that PKC98E is located in the dendritic cilia of ORNs where olfactory signaling occurs. Because PKC98E is required for dorsal-ventral patterning in the embryo, mutants were lethal [104]. This can explain why this kinase was not identified in genetic screens [105]. Subsequently, a conditional PKC98E knockout specific to mature olfactory neurons showed a similar level of reduction in olfactory responses. Immunostaining using Orco-S289-phosphospecific antiserum revealed a striking reduction in basal Orco S289 phosphorylation, but Orco protein levels and localization were unaffected [103]. A constitutively active PKC98E that lacked the C1 regulatory domain and contained a CAAX signal at the C-terminus to tether the enzyme to the membrane rescued the Orco S289 phosphorylation in PKC98E conditional mutants and rescued the defective olfactory response to cVA [103]. Together, these studies unveiled a molecular mechanism underlying the long-term desensitization in *D. melanogaster* (Figure 2). 

## 4. Future Directions 

### 4.1. Factors Activating PKC98E

PKC98E belongs to nPKCs, which lack a calcium-binding domain. Therefore, the activity of PKC98E is not subject to calcium regulation but rather hinges on diacylglycerol (DAG) and phosphatidylserine (PS) [106,107,108]. Interestingly, the PS flippase dATP8B, which flips the PS from the outer leaflet of a cell membrane to the inner leaflet, is necessary for normal olfactory responses in *Drosophila* [109,110]. Flies that were defective in dATP8B only showed residual response to cVA above 1%, a phenotype that is reminiscent of the responses of the Orco S289A mutant [110]. The location of dATP8B is confined to the dendritic cilia of ORNs where PKC98E is present, suggesting that they may function in the same pathway. The questions are whether the activity or localization of PKC98E is affected by the presence of PS and whether PS levels change with odorant stimulation. In the future, using the *PKC98E* allele PKC98E^CAAX^ to restore the defective odorant response of dATP8B will answer this question [103]. The dominant kinase should bypass the need for dATP8b to maintain S289 phosphorylation. If true, considering that the phosphorylation of Orco S289A is dynamically regulated by odorant exposure, the PS composition in the membrane of ORNs should be dynamically regulated by odorant exposure. 

### 4.2. Phosphatase Dephosphorylating Orco S289

We propose that the dephosphorylation of Orco S289 results in the desensitization of olfactory receptors [77,79]. Yet, the phosphatase(s) responsible for this dephosphorylation are still unknown. Systematic screening of the defective dephosphorylation of Orco S289 by immunostaining, using the phosphorylation-specific antibody in the genetic background of each phosphatase knockdown, may finally identify the phosphatase. Alternatively, using SSR to examine the defective desensitization when each phosphatase gene is knocked down may be conducive to finding this phosphatase.

### 4.3. Molecular Mechanism of Desensitization of ORNs

At least 20 kinds of kinases are expressed in the *D. melanogaster* antennae [104]. PKC98E is the major kinase phosphorylating Orco and was implicated in the long-term desensitization of ORNs [104]. It is well-known that the short-term desensitization of *Drosophila* ORNs is triggered by the influx of calcium [88]. Are any of them responsible for short-term desensitization? PKC53E and PKC delta were shown to phosphorylate Orco to enhance sensitivity [111]. Although they are not needed for long-term desensitization [103], they may participate in short-term desensitization. Apart from phosphorylation sites, a calmodulin-binding site, SAIKYWVER, was found on the second intracellular loop of *Drosophila* Orco [112]. This site is necessary for the sensitization of ORs by repeated stimulation of subthreshold odorants [112] and the trafficking of OR-Orco from the soma to outer dendritic cilia [113]. Despite that, it is probable that this site is also involved in desensitization triggered by continuous stimulation or repeated pulse stimulation of high doses of odorants.

## Figures and Tables

**Figure 1 insects-13-00354-f001:**
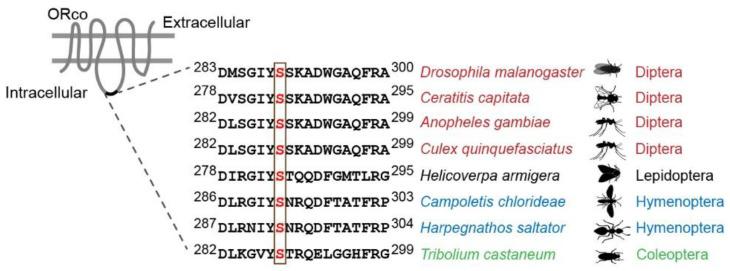
Conservation of the S289 phosphorylation site of Orco across different insect species. The amino acid sequences of Orco were downloaded from the National Center for Biotechnology Information (NCBI): flies *D. melanogaster* (NP_524235.2) and *Ceratitis capitata* (NP_001266301.1); mosquitoes *Anopheles gambiae* (XP_041762121.1) and *Culex quinquefasciatus* (ABB29301.1); moth *Helicoverpa armigera* (ADQ13177.1); parasitoid *Campoletis chlorideae* (AKO69815.1); ant: *Harpegnathos saltator* (XP_011139767.1); beetle *Tribolium castaneum* (CAM84014.1).

**Figure 2 insects-13-00354-f002:**
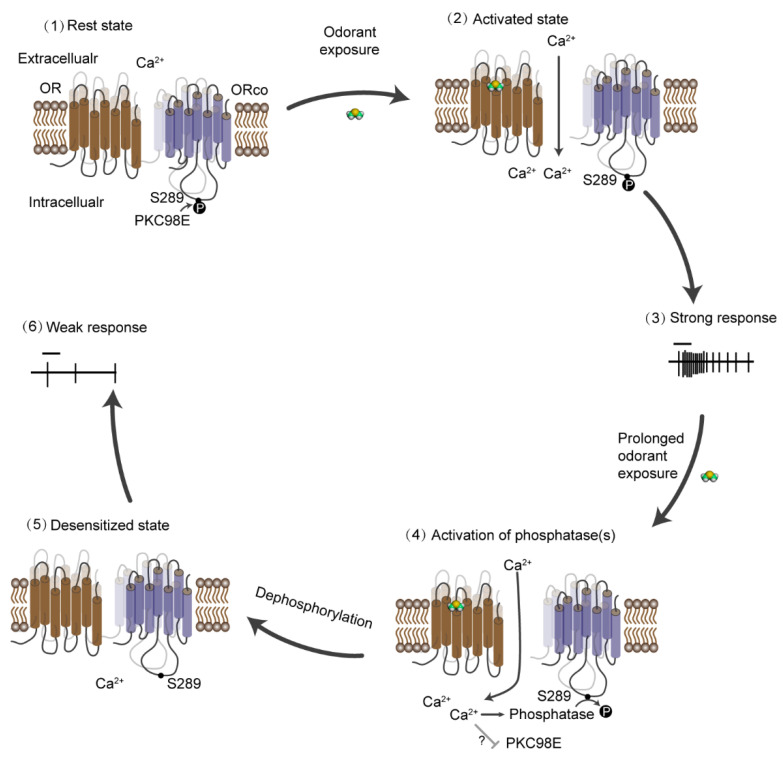
Schematic diagram of the process of long-term desensitization in *D. melanogaster*. (**1**) At rest state, S289 is phosphorylated by PKC98E, and the olfactory receptor channel is sensitive to the odorant stimulation. (**2**) The ligands bind to the receptors and open the channel, leading to the influx of calcium and (**3**) the generation of strong action potentials. (**4**) When the presence of stimuli is prolonged, the accumulating intracellular calcium activates unknown phosphatase(s) that dephosphorylate Orco S289. (**5**) This dephosphorylation gradually desensitizes the odorant receptor channels, which is reflected by the inhibited electrophysiological response to odorant stimuli (**6**).

## Data Availability

Not applicable.

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
