# Peer review of "Time-Dependent Odorant Sensitivity Modulation in Insects"

_insects, 2022, doi:10.3390/insects13040354_

Round 1

Reviewer 1 Report

This is a useful review concerning insect olfactory transduction, particularly desensitization. Overall, it was straightforward to follow and well-referenced. Sections 2, 3.2, and 4 were particularly well written.

In Section 3, the discussion of vertebrate desensitization is useful. However, the portion from line 195-209 is difficult to follow, as it brings up conflicting data on whether insect ORs are GPCRs or activate signal transduction cascades without tying this directly to desensitization, the topic at hand. This information could be be removed or greatly simplified or perhaps moved to the introduction (section 1).

The discussion in section 3.1 feels a bit scattered. The first paragraph concerns OBPs and EST6. The second Ca2+ mentioned, cGMP, and trp. Reorganization of this section should be considered. It may be helpful to consider putting in place a model framework- ex. 1) does desensitization to prolonged odor exposure happen at the intrinsic receptor level (receptor conformational change- should be present when expressed in heterologous cells), 2) is desensitization induced by a signal transduction cascade (Ca2+, cGMP, trp, etc), or 3) does desensitization involve changes in odorant availability (OBPs, ODEs).

Additionally, it is unclear why the section switches focus from desensitization to deactivation, two distinct pharmacological processes. 

Additional minor comments:

1) Line 71, add reference #94 (Larter et al)

2) Line 88, use the standard Flybase name for Orco (not ORco)

3) Section 2.2, cite other references demonstrating desensitization in Drosophila such as De Bruyne et al J Neuro 1999, Deshpande et al J Neurobio 2000, Baldwin et al Scientific Reports 2021

4) Line 197, should clarify "involved" to "involved in downstream modulatory processes" or something along this line

5) Section 3.1, incorporate work from Nagel et al Nature Neuroscience 2011

Author Response

This is a useful review concerning insect olfactory transduction, particularly desensitization. Overall, it was straightforward to follow and well-referenced. Sections 2, 3.2, and 4 were particularly well written.

Thanks for this positive comment. We have now revised our manuscript accordingly. Please find our responses below.

In Section 3, the discussion of vertebrate desensitization is useful. However, the portion from line 195-209 is difficult to follow, as it brings up conflicting data on whether insect ORs are GPCRs or activate signal transduction cascades without tying this directly to desensitization, the topic at hand. This information could be removed or greatly simplified or perhaps moved to the introduction (section 1).

For clarity, we have moved this portion of discussion to the Introduction (Line 91-106).

The discussion in section 3.1 feels a bit scattered. The first paragraph concerns OBPs and EST6. The second Ca2+ mentioned, cGMP, and trp. Reorganization of this section should be considered. It may be helpful to consider putting in place a model framework- ex. 1) does desensitization to prolonged odor exposure happen at the intrinsic receptor level (receptor conformational change- should be present when expressed in heterologous cells), 2) is desensitization induced by a signal transduction cascade (Ca2+, cGMP, trp, etc), or 3) does desensitization involve changes in odorant availability (OBPs, ODEs).

We agree with the reviewer that this part of discussion lacks logic. We have reorganized it and made appropriate changes to make it more coherent as per the reviewer’s suggestions (see the revised section 3.1).

Additionally, it is unclear why the section switches focus from desensitization to deactivation, two distinct pharmacological processes. 

We believe that receptor deactivation is part of olfaction desensitization. We have made this point clear in the manuscript (Line 269-272).

Additional minor comments:

  • Line 71, add reference #94 (Larter et al)

Done

  • Line 88, use the standard Flybase name for Orco (not ORco)

Now we have adopted the writing of Orco throughout the manuscript.

  • Section 2.2, cite other references demonstrating desensitization in Drosophila such as De Bruyne et al J Neuro 1999, Deshpande et al J Neurobio 2000, Baldwin et al Scientific Reports 2021

We have cited these references. Thanks.

4) Line 197, should clarify "involved" to "involved in downstream modulatory processes" or something along this line

We have made the change accordingly. Moreover, we have moved this part of discussion to the Introduction section.

5) Section 3.1, incorporate work from Nagel et al Nature Neuroscience 2011

Done

Reviewer 2 Report

In this manuscript Gun and Smith present a thorough review on olfactory adaptation at the ORN level, with a focus on the molecular mechanisms behind long-term and short-term desensitization. I think the manuscript is well written and very engaging, with a really good flow of information. 

I have 3 minor comments regarding the manuscript:

1.- Line 62- the authors should cite the work of Gomez-Diaz et al, PLoS Biol. 2013;11(4):e1001546 acknowledging the controversy of LUSH as an activator of Ors.

2.- Some of the literature reviewed on descriptions of the desensitization phenomena do not differentiate between mechanisms allocated in the sensilla versus other OSN compartments. Maybe the authors could briefly mention OSN adaptation mechanisms like the ones described by Murmu et al in BMC Neurosci. 2011;12:105 before focusing on pure odorant-reception mechanisms

3.- I found several typos in the manuscript, please revise and spell-check (example: lines 42 odorant receptor(s); 117 results indicate(s); 222 has a significantly higher response (s)

My congratulations to the authors, I enjoyed this review very much

Author Response

In this manuscript Gun and Smith present a thorough review on olfactory adaptation at the ORN level, with a focus on the molecular mechanisms behind long-term and short-term desensitization. I think the manuscript is well written and very engaging, with a really good flow of information. 

We thank and appreciate the positive comments from the reviewer. We have revised our manuscript accordingly. Please find the responses below.  

I have 3 minor comments regarding the manuscript:

  1. Line 62- the authors should cite the work of Gomez-Diaz et al, PLoS Biol. 2013;11(4):e1001546 acknowledging the controversy of LUSH as an activator of Ors.

We do not think that the work of Gomez-Diaz et al. (2013) supports the role of OBPs as Or activators. Therefore, we are inclined to leave it out at this stage.

  1. Some of the literature reviewed on descriptions of the desensitization phenomena do not differentiate between mechanisms allocated in the sensilla versus other OSN compartments. Maybe the authors could briefly mention OSN adaptation mechanisms like the ones described by Murmu et al in BMC Neurosci. 2011;12:105 before focusing on pure odorant-reception mechanisms

Thanks for this suggestion. We have cited this reference and briefly touched on the mechanisms involving presynaptic inhibition (lines 140-146).

  1. I found several typos in the manuscript, please revise and spell-check (example: lines 42 odorant receptor(s); 117 results indicate(s); 222 has a significantly higher response (s)

Done